# Change of Sleep Stage during Gastroesophageal Reflux in Infants

**DOI:** 10.3390/children10050836

**Published:** 2023-05-04

**Authors:** Angeliki Pappa, Moritz Muschaweck, Tobias G. Wenzl

**Affiliations:** Klinik für Kinder-und Jugendmedizin, Universitätsklinikum der RWTH Aachen, Pauwelsstr. 30, 52074 Aachen, Germany

**Keywords:** sleep stage change, gastroesophageal reflux, temporal association, infant

## Abstract

Introduction: This study intended to explore the existence of a temporal association of changes of sleep stage and gastroesophageal reflux (GER) in infants. Materials and Methods: Documentation of sleep stage and GER was conducted via the use of synchronized polygraphic recording combined with impedance-pH-metry in 15 infants. The total recording-time (Rt) was divided into GER-“window-time” (five seconds before and after the onset of a GER episode), “remaining GER time”, and “GER-free time”, and analyzed for changes of sleep stage. Results: a total of 462 GER episodes were identified during Rt (151.1 h) in all infants. During 1.3 h of window-time; 61 changes of sleep stage (47/h); during 5.9 h of Remaining GER-time, 139 changes of sleep stage (24/h); and during 143.9 h of GER-free time, 4087 changes of sleep stage (28/h) were documented. Change of sleep stage was strongly associated with the onset of GER (*p* < 0.02 and *p* < 0.05, respectively). Conclusions: There is a strong temporal association between sleep irregularities, i.e., changes of sleep and episodes of GER in infants. When dealing with disturbed sleep in infants, GER should be considered by caregivers.

## 1. Introduction

A relationship between gastroesophageal reflux (GER) and various extraesophageal symptoms has been postulated [1]. These symptoms are nonspecific such as cardiorespiratory and neuromotor symptoms, irritability and vomiting. They can also relate to non-reflux situations such as chronic lung disease, intra-cerebral bleeding and apnea of premature infants [2,3]. These findings are common, more than half of the population younger than three months show signs of GER, and the observation of GER disease in neonatal intensive care units (NICU) infants approximates 10% (20).The recording of multiple intraluminal impedance and pH (MII-pH) on one catheter is currently the accepted first-line diagnostic tool for this situation in children of all age groups. Detecting not only acid reflux but reflux in general can provide more information than only pH monitoring. Clinical circumstances, in which non-acid or weakly acid reflux may be causal, requires such a tool [4,5,6].

Disturbed sleep is commonly observed in infants with gastroesophageal reflux. A pre-reflux alteration in the autonomic nervous system (ANS), as judged by change of heart rate, has been observed as a factor for the mechanism of reflux in neonates. Heart rate variability, and time- and frequency-domain parameters, are influenced by the vigilance state. GER events show a distinct pattern, depending on state of vigilance (wakefulness, active sleep, quiet sleep) [2,7]. 

This study intended to investigate a proposed temporal association of changes of sleep stages, as a denominator of disturbed sleep, and GER in infants with combined MII-pH-metry and synchronized polygraphic recording.

## 2. Materials and Methods

Fifteen infants (mean age 97 ± 52 days; 8 female, 7 male) with recurrent regurgitation were examined over approximately 10 h each. Medication to control GER was not applied during the study. The infants received their regular milk formula and were placed supine postprandially.

For recording and analysis of MII-pH, the standard protocol was used, following the ESPGHAN (European Society of Pediatric Gastroenterology, Hepatology and Nutrition, Geneva, Switzerland) EURO-PIG (European Pediatric Impedance Group, Aachen, Germany) guideline, as described previously [6,8,9].

For sleep stage documentation, polygraphic recording was performed with a standard sleep lab system (ALICE; Heinen & Löwenstein, Bad Ems, Germany) at the bedside. Recorded parameters included EEG, periocular EMG, heart rate, respiratory rate, and gross motor activity (by actinometer and infrared sensor). The subdivision of sleep stages was into wakefulness, active sleep, and quiet sleep. 

The definition of active sleep included: predominant theta-activity with low amplitude (EEG), eyes closed, rapid eye movement (REM), irregular breathing, fluctuation of heart rate, and phasic movement. 

The definition of quiet sleep included: predominant delta-activity with high amplitude (EEG), eyes closed, no REM, regular breathing, little or no fluctuation of heart rate, and little or no movement [10].

All polygraphic parameters were recorded simultaneously and picture-in-picture, together with the MII-pH signal, with a recording system at the bedside.

MII-pH data were visually analyzed for the typical impedance pattern of GER, indicated by a retrograde esophageal volume flow [6]. GER diagnosis was defined by the typical pattern being detected in the esophageal impedance tracings. The pH-nadir for each GER was documented.

The total analyzable time of polygraphic- and MII-pH recording in all patients was defined as “recording-time” (Rt). GER episodes that occurred during Rt underwent further evaluation. Per definition, Rt was divided into “window-time” (five seconds before and after the onset of a GER) (Figure 1), “Remaining GER-time” and “GER-free time”.

All polygraphic recordings were analyzed, and all changes of sleep stage (i.e., change from wakefulness to active sleep or quiet sleep, from active sleep to quiet sleep or wakefulness and from quiet sleep to active sleep or wakefulness) were documented.

Rt was analyzed for frequency of changes of sleep stage during the different time periods (window-time; remaining GER-time; and GER-free time).

The paired Wilcoxon test was used for statistical analysis. Significance was established where there was a *p* value of <0.05.

The study protocol was approved by the Ethics Committee of the Medical Faculty, RWTH Aachen, Germany. Before commencing any evaluation of an infant, written informed consent was obtained from the parents.

## 3. Results

During a total analyzable recording time (Rt) of 151.1 h, 462 GER episodes were registered in the 15 infants. This resulted in a total window-time of 1.3 h (0.9% of Rt). Remaining GER-time was 5.9 h (3.9%); GER-free time was 143.9 h (95.2%). A total of 65 (14%) GER episodes were acidic (pH < 4).

During Rt, 4287 changes of sleep stage were registered. During total window-time (1.3 h), 61 changes of sleep stage occurred (47 per hour); during the remaining GER-time (5.9 h), 139 changes of sleep stage occurred (24 per hour); and during GER-free time (143.9 h), 4087 changes of sleep stage occurred (28 per hour) (Figure 2).

When compared to the remaining GER-time and GER-free time, window-time (i.e., the onset of a GER episode) was significantly associated with the manifestation of a change of sleep stage (*p* < 0.02 and *p* < 0.05, respectively), irrespective of the nature of this change. There was a slight trend towards an arousal reaction (i.e., a change from quiet sleep to active sleep or wakefulness, or a change from active sleep to wakefulness), without reaching statistical significance. Likewise, there was no statistical difference regarding the pH of the GER.

## 4. Discussion

Aspects of GER, and its association with different extraesophageal symptoms, previously underwent evaluation in children [11,12,13]. However, the potential influence of GER on sleep stage changes in infants has so far not been scientifically validated with the combined use of synchronized MII-pH and polygraphic recording. The combination of these tools can help to identify cardiorespiratory symptoms distinctly during sleep, and to enable further investigation of differentiating GER disease from other etiologies [2,7]. Later studies have tested the pattern of GER events in wakefulness and sleep states, and described an increased acid reflux index (>7%) in the wake-state compared to the sleep-state.

In this study, an association of change of sleep stage and gastroesophageal reflux was found in infants. During the cumulative time surrounding the onset of a GER episode, 47 changes of sleep stage occurred per hour, whereas during the cumulative remaining duration of all GER episodes, and the recording-time without a GER episode, only 24 and 28 changes of sleep stage occurred per hour, respectively. This difference was statistically significant, though independent of the nature of the change of sleep stage and the pH of the reflux. In their study, Djeddi et al. examined the distribution of GER events and found more events during active sleep than during quiet sleep [7]. Using a big data approach could bring new insights into subsequent studies. Ehsan et al. used large-scale aggregate data to examine sleep-disordered breathing in 6429 infants; one of the most common comorbid diagnoses was GER [14].

These findings should not be considered a specific pathology in general, which is also true for uncomplicated GER in infants, as we found a considerable amount of changes of sleep stage towards “deeper” sleep during the onset of a GER episode. Gastroesophageal reflux is a common finding in infants and probably constitutes a physiological phenomenon that may trigger a vast spectrum of extraesophageal reactions [15,16]. Unexplained distress can be specified via assessment using the Face, Legs, Activity, Cry, Consolability (FLACC) scale, and by using synchronized MII-pH-metry and polysomnography [16].

One could speculate that in some infants, a resolved GER episode may allow the onset of rest and a change to a deeper sleep stage, as we could document in some patients.

In contrast, in some infants, an arousal reaction is triggered by the onset of a GER episode, possibly to increase vigilance and facilitate reflux clearance. These phenomena could not be differentiated with the chosen study design and the small number of patients included. In further studies, the integrative physiological approach, under optimal feeding and crib-side testing conditions, could be useful [17,18].

However, as shown here, sleep irregularities in general are clearly associated with reflux episodes, and sleep disturbances associated with GER should therefore be considered in the differential diagnosis in infants [2,7,19].

Further investigation is needed for identifying risk factors and planning possible therapeutic strategies [20].

## Figures and Tables

**Figure 1 children-10-00836-f001:**
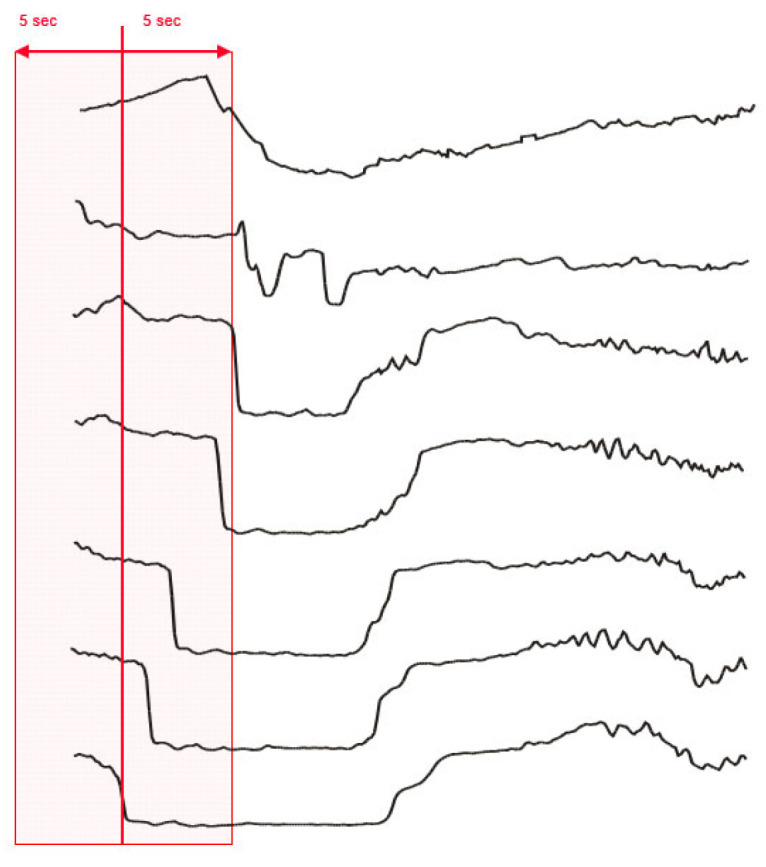
Window-time. “Window-time”, defined as five seconds before and after the onset of a GER.

**Figure 2 children-10-00836-f002:**
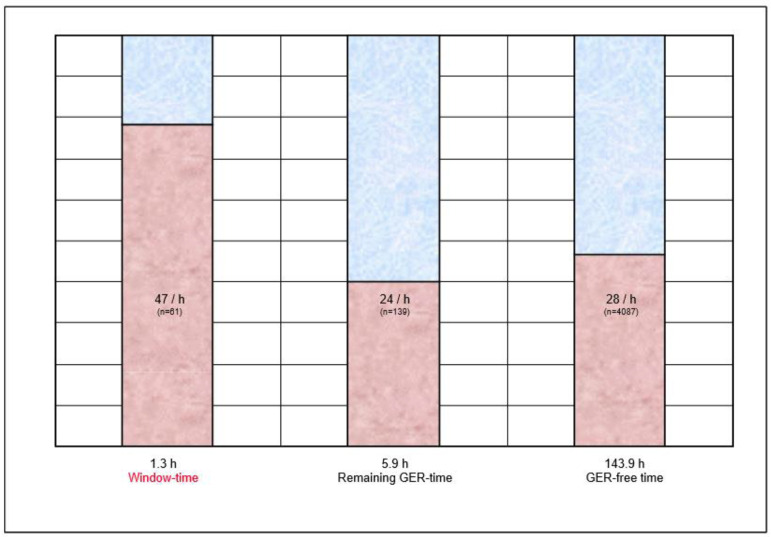
Association of change of sleep stage and GER. Window-time = 5 s before/after the beginning of a GER episode; remaining GER-time = GER-duration minus 5 s; GER-free time = recording time without GER. *n* = number of changes of sleep stage.

## Data Availability

The data presented in this study are available on request from the corresponding author. The data are not publicly available due to ethical restrictions.

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
