# Peer review of "Change of Sleep Stage during Gastroesophageal Reflux in Infants"

_children, 2023, doi:10.3390/children10050836_

Round 1

Reviewer 1 Report

The purpose of the study is good. However, the case series use only 15 patients and therefore cannot give valid results. The description of the methods and results is not easily understood.

I have noticed many self-citations

Author Response

We thank Reviewer 1 very much for commenting on our manuscript, especially for giving the purpose of our study the attribute of being good.

We tried to relate to all comments and respond in detail:

Unfortunately, at this point the research design and the number of patients included could not be changed any more. The validity of our results due to the small number of patients was commented on in the Discussion. We tried to improve the description of the methods and the presentation of the results for better understanding. The English language was rechecked throughout the manuscript.

The self-citations are mostly a result of trying to reduce redundancy in our manuscript for the readers, as some of the cited standard protocols are also authored by one of the authors of this manuscript.

We hope we could provide sufficent comment to all issues raised by the Reviewer. Again, thank you very much.

Reviewer 2 Report

Dear Authors, this paper about change of sleep stage during gastroesophageal reflux in infants is very interesting and is going to be very important both to the scientific community and to dental workers.

Some small issues need to be solved before its final publication in the journal.

Abstract: please dive abstract into: introduction, materials and methods, results, conclusions. In this way the abstract will be much easier to read and understood by readers.

Introduction: this part in really important and it helps readers to deep into the subject before passing to the materials and methods part. In this article, introduction is really short. It needs to be lengthened: please add 2 more chapters: 1 about hot to prevent erosive dental problems in children affected by gastroesophageal reflux disease: this paper may help: Ludovichetti, F.S.; Zambon, G.; Cimolai, M.; Gallo, M.; Signoriello, A.G.; Pezzato, L.; Bertolini, R.; Mazzoleni, S. Efficacy of Two Toothpaste in Preventing Tooth Erosive Lesions Associated with Gastroesophageal Reflux Disease. Appl.Sci.2022,12,1023. https:// doi.org/10.3390/app12031023

And a second chapter about gastroesophageal reflux disease and aero digestive disorders in general: 

  1. Maqbool, A.; Ryan, M.J. Gastroesophageal Reflux Disease and Aerodigestive Disorders. Curr. Probl. Pediatr. Adolesc. Health Care

    201848, 85–98

Materials and methods: instead of patients and methods, write materials and methods.

Discussion: this part, once again, is too short, please enhance your discussion. Describe more in details your results and compare them with the available literature: 

Ludovichetti FS, Signoriello AG, Girotto L, Del Dot L, Piovan S, Mazzoleni S. Oro-dental lesions in pediatric patients with celiac disease: an observational retrospective clinical study. Rev Esp Enferm Dig. 2022 Nov;114(11):654-659.

Howard JP, Howard LJ, Geraghty J, Leven AJ, Ashley M. Gastrointestinal conditions related to tooth wear. Br Dent J. 2023 Mar;234(6):451-454. doi: 10.1038/s41415-023-5677-0. Epub 2023 Mar 24.

Author Response

We thank Reviewer 2 very much for commenting on our manuscript, especially for giving it the attributes of being very interesting and very important, and finding the research design appropriate, the methods adequately described, the results clearly presented and the conclusions supported by te results. We tried to relate to all comments and respond in detail:

The English language was rechecked throughout the manuscript.

We divided the Abstract according to the Reviewers suggestion.

We tried to improve the introduction according to the Reviewers comments. However, as none of the authors has sufficient expertise in dentistry, we would prefer not to enter this field here. We agree that airway disorders associated with GER are a very important topic in infants, however they were not part of this study. They have been covered by one of the authors in previous publications, and were therefore not included in the current manuscript, moreover to prevent additional self citations. In accordance with the Special Issue Editor, we purposely kept the manuscript short and concise, as we hoped by this to find more approval by readers.

As suggested by the Reviewer we renamed the Patients and Methods section to Materials and Metods.

The suggested extension of the Discussion regarding oro-dental lesions, as suggested by the Reviewer is unfortunately, as described above, outside the expertise of the authors.

We hope we could provide sufficent comment to all issues raised by the Reviewer. Again, thank you very much.

Round 2

Reviewer 1 Report

Accept in present form